# Bacterial Endosymbiont Diversity among *Bemisia tabaci* (Hemiptera: Aleyrodidae) Populations in Florida

**DOI:** 10.3390/insects11030179

**Published:** 2020-03-11

**Authors:** Bruno Rossitto De Marchi, Hugh A. Smith

**Affiliations:** Gulf Coast Research and Education Center, University of Florida, Wimauma, FL 33598, USA; hughasmith@ufl.edu

**Keywords:** MEAM1, *Rickettsia*, *Hamiltonella*, facultative

## Abstract

The sweetpotato whitefly, *Bemisia tabaci* (Hemiptera: Aleyrodidae), is a pest of many economically important agricultural crops and a vector of plant viruses. *Bemisia tabaci* harbors facultative endosymbiont species that have been implicated in pest status, including tolerance to insecticides, virus transmission efficiency and tolerance to high-temperatures. The facultative endosymbionts reported in *B. tabaci* include *Arsenophonus*, *Hamiltonella*, *Wolbachia*, *Cardinium*, *Fritschea* and *Rickettsia*. We collected whitefly populations from weed and crop hosts in south Florida and identified the whitefly species as well as the facultative endosymbionts present in these populations by molecular analysis. In addition, a phylogenetic analysis of whiteflies and their endosymbionts was performed. The only facultative endosymbionts found among the *B. tabaci* populations collected in Florida were *Hamiltonella* and *Rickettsia*. The phylogenetic analysis revealed the low genetic diversity of whiteflies and their endosymbionts. Additionally, the phylogenetic tree clustered *Rickettsia* from Florida in the R1 genetic group. The results will aid to understand the role of the bacterial endosymbionts in the whitefly host.

## 1. Introduction

The whitefly, *Bemisia tabaci* (Hemiptera: Aleyrodidae), is a pest of many agricultural crops and a successful vector of over 300 plant virus species including *Begomovirus*, *Crinivirus*, *Carlavirus*, *Torradovirus* and *Ipomovirus* [1,2]. Currently, *B. tabaci* is considered a species complex based on the mitochondrial cytochrome oxidase I (mtCOI) gene [3]. The different genetic groups are indistinguishable morphologically but several other characteristics differ among them [4,5] such as plant host preference, fecundity, ability to transmit viruses [6,7], dispersal and resistance to insecticides [8]. In the United States, three species of the *B. tabaci* complex have been reported: the Middle East-Asia Minor 1 (MEAM1), also known as B biotype; the Mediterranean (MED), also named as Q biotype; and the indigenous species New World 1 (NW1) or A biotype [9]. *Bemisia tabaci* MEAM1 is the primary pest of tomato (*Solanum lycopersicum* L.) in Florida and recent surveys indicate that MED species has been detected on landscape plants in residential areas [10]. Florida is among the main producers of fresh market tomatoes in the United States, with over 27,000 acres harvested in 2018 at a value of $344 million [11]. In addition to tomatoes, *B. tabaci* is a major pest of other important vegetable crops in Florida such as peppers, squash, cucumber, beans, eggplant, watermelon and cabbage. It can also heavily attack potato, peanut, soybean and cotton and ornamentals including poinsettia, hibiscus and chrysanthemum [12]. The *B. tabaci* species complex carries an obligatory primary endosymbiont called *Candidatus Portiera aleyrodidarum* [13]. In addition, whiteflies may harbor facultative endosymbiont species that have been implicated in pest status, including tolerance to insecticides [14], virus transmission efficiency [15,16] and tolerance to high-temperatures [17]. The facultative endosymbionts reported in *B. tabaci* include *Hamiltonella* and *Arsenophonus* [13], *Wolbachia* [18], *Cardinium* [19], *Fritschea* [20] and *Rickettsia* [21]. Previous studies have revealed a complex genetic divergence of the different endosymbionts within *B. tabaci* [9]. The infection dynamics of facultative endosymbionts in *B. tabaci* are usually associated with whitefly species, gender, host plants and geographical locations [22]. However, there is a lack of information about the facultative endosymbiont diversity in whitefly populations of Florida. In this study, we collected whiteflies from weed and crop hosts in south Florida and identified the whitefly species as well as the facultative endosymbionts by molecular analysis. Additionally, a phylogenetic analysis of the resulted sequenced data was performed. The goal of this study was to report the presence and to infer the diversity among the whitefly facultative endosymbionts in populations of Florida. Knowledge regarding the facultative endosymbionts can contribute to understanding epidemics of whitefly-transmitted viruses and be helpful in resistance monitoring studies. The results obtained will supplement further studies to determine relations between facultative endosymbionts and pest status.

## 2. Material and Methods

### 2.1. Field Collection

Populations of *B. tabaci* were collected from five south Florida counties in 2019: Hendry, Collier, Manatee, DeSoto and Miami-Dade (Figure 1). Whitefly adults from different host plants, including tomato, eggplant, watermelon, sweetpotato, hibiscus, *Parthenium* sp., *Euphorbia* sp., *Amaranthus* sp. and *Bidens* sp., were aspirated into vials using a hand-held aspirator and then transferred to cloth covered cages with cotton (*Gossypium hirsutum* L) seedlings. The cages were brought back to the University of Florida’s Gulf Coast Research and Education Center in Balm, FL, and maintained on cotton plants in growth rooms at 27 °C (±2 °C), 50% to 75% RH and 14:10 (L:D) photoperiod. Ten whiteflies per population were aspirated and transferred to vials containing 95% EtOH and stored at −20 °C for molecular analysis.

Two laboratory colonies, one of *B. tabaci* MEAM1, the other of *B. tabaci* MED, were analyzed in addition to the field populations collected in 2019. The MEAM1 colony was established in the 1990s from whiteflies collected near Bradenton, FL. The MED colony was established in July 2017 from whiteflies collected from hibiscus (*Hibiscus rosa-sinensis* L) in Palm Beach County, FL. Each colony was maintained in a separate growth room under the same conditions as the field populations.

### 2.2. Whitefly Identification

Total nucleic acids were extracted from each individual whitefly, following a modified Chelex protocol [23]. Ten insects for each population were individually tested. *Bemisia tabaci* adults were homogenized in 40 μL of 5% Chelex solution in a 1.5 mL tube. The tube was vortexed for a few seconds and then incubated at 56 °C for 15 min and at 99 °C for 8 min. After centrifugation at 13,000 rpm for 5 min, the supernatant was collected and used as a template for the PCR amplification. The PCR to differentiative MEAM1 from MED was carried out using the primer pair Bem23F (5’-CGGAGCTTGCGCCTTAGTC-3’) and Bem23R (5’-CGGCTTTATCATAGCTCTCGT-3’), which amplifies a microsatellite locus of about 200 bp and 400 bp for MEAM1 and MED [24,25,26], respectively. The PCR product was visualized by electrophoresis in 2% agarose gel stained with Gel Red. Later, at least one sample for each population was used for a PCR with the generic insect primers C1-J-2195 and TL2-N-3014 that amplify a fragment of the mtCOI [26] followed by PCR purification and Sanger sequencing of the fragment (GENEWIZ, South Plainfield, NJ, USA). The nucleotide sequences were analyzed and compared with those present in the GenBank database using BLAST tools (http://blast.ncbi.nlm.nih.gov/Blast).

### 2.3. Facultative Endosymbionts Identification

The identification of facultative endosymbionts was performed using the same DNA that was used previously for the whitefly species identification. The PCR was carried out with previously described genus-specific primers targeting the 16S or the 23S genes for the identification of *Hamiltonella* [18], *Rickettsia* [21], *Wolbachia* [27], *Cardinium* [19], *Arsenophonus* [28] and *Fritschea* [20]. Later, amplicons of some representative populations were selected for DNA sequencing (GENEWIZ, South Plainfield, NJ, USA). The nucleotide sequences were analyzed and compared with those present in the GenBank database using BLAST tools (http://blast.ncbi.nlm.nih.gov/Blast).

### 2.4. Phylogenetic Analysis

The Bayesian analyses were conducted using Mr. Bayes v. 3.2.278. They were run for 30 million generations with sampling every 1000 generations. Each analysis consisted of four independent runs, each utilizing four coupled Markov chains. The run convergence was monitored by finding the plateau in the likelihood scores (standard deviation of split frequencies <0.0015). It was discarded in the first 25% of each run. The resulted trees were edited and rooted using FigTree v1.4.2.

The whitefly phylogenetic analysis was performed using the 15 *B. tabaci* mtCOI sequences obtained in this study added to a global *B. tabaci* mtCOI dataset [29] totalizing 774 sequences. In addition, phylogenetic analyses of the Partial 16S rDNA gene were carried out for the facultative endosymbionts *Hamiltonella* and *Rickettsia*. Multiple sequence alignments were prepared using MAFFT77 within the Geneious 9.1.5 software.

## 3. Results

The prevalent facultative endosymbionts pattern observed throughout Florida was *Rickettsia* in co-infection with *Hamiltonella* (Figure 2).

This pattern was found in all the regions collected and was the most common pattern in Homestead. In the Myakka City/Arcadia area, most of the specimens found were only harboring *Hamiltonella*. We also identified the set of endosymbionts for the laboratory colonies, MEAM1 (80% *Hamiltonella* + 70% *Rickettsia*) and MED (100% *Rickettsia* + 50% *Arsenophonus*), as references. The MED colony was the only population harboring the facultative endosymbionts *Arsenophonus*.

The *B. tabaci* mtCOI sequence alignment was 625 bp in length. The *Hamiltonella* alignment consisted of a total of 44 sequences and 746 bp in length. The *Rickettsia* alignment consisted of a total of 22 sequences and 628 bp in length. The mtCOI phylogenetic analysis grouped all the *B. tabaci* field populations from Florida in the MEAM1 clade. No variability was observed among the populations collected (Figure 3).

Additionally, low diversity was observed for the 16S gene rDNA of *Hamiltonella* as all the nucleotide sequences were grouped into one major cluster (Figure 4A). The phylogenetic analysis of the 16S gene rDNA of *Rickettsia* clustered the Florida populations in the R1 genetic group (Figure 4B).

## 4. Discussion

The survey was conducted in five different counties, and we analyzed 27 field populations from at least seven different host plants, including the cultivated crops tomato, sweetpotato and hibiscus, as well as weeds, including *Parthenium* sp., *Euphorbia* sp. and *Amaranthus* sp. (Table 1). Among the field populations, all the specimens were identified as *B. tabaci* MEAM1 species and the only facultative endosymbionts found were *Hamiltonella* and *Rickettsia* (Figure 1).

The high rates of *Hamiltonella* in some populations collected from tomatoes is a concern as this endosymbiont has a role in the transmission of the begomovirus, *Tomato yellow leaf curl virus* (TYLCV), which causes yellow leaf curl disease in tomato. *Hamiltonella* encodes a GroEL chaperonin homologue protein that safeguards the begomovirus particles in the hemolymph of the whitefly vector [15]. This disease is a major problem in Florida and is estimated to cause tens of millions of dollars in losses for the tomato crop [30].

The facultative endosymbionts *Rickettsia* was also found in different rates across Florida. Several populations were found with rates lower than 30% (i.e., Populations #8, #9, #12, #13, #18, #22, #24, #25, #33, #34, #42, #43). A previous study has associated the presence of *Rickettsia* to increased insecticide susceptibility to acetamiprid, thiamethoxam, spiromesifen and pyriproxyfen in whiteflies [14]. Thus, populations with low incidences of *Rickettsia* might be a factor to be observed in a resistance monitoring study.

The first collection in the Myakka City/Arcadia area was carried out in February 2019, and a different set of endosymbionts was found in two populations from tomato fields located less than 3 km apart. Population #6 harbored *Rickettsia* (88%) + *Hamiltonella* (66%). Population #8 harbored only *Hamiltonella* (90%). Three months later, we sampled the same tomato field where Population #6 was found and the facultative endosymbionts *Rickettsia* was no longer detected. The new sampling was identified as Population #24 (10% *Hamiltonella*). A shift in the dynamics of facultative endosymbionts in whiteflies has been reported before in several studies. In Arizona, USA, a decline was reported in *Rickettsia* infection from ~95% in 2011 to 36% in 2017 in a six-year survey of *B. tabaci* MEAM1 in cotton [31]. In Brazil, the frequencies of *Hamiltonella* and *Rickettsia* remained very similar in *B. tabaci* MEAM1 species from 2015 to 2017. However, the same study reported the increasing of infection of *Hamiltonella*, *Rickettsia* and *Wolbachia* in MED species individuals from 2015 to 2017 [32]. In laboratory conditions, *Rickettsia* is vertically transmitted by female whiteflies at high rates ranging from 98%–99% [31,33]. Therefore, some unknown factors in the field must be responsible for the decline in *Rickettsia* infection. There are evidences that the decline might be driven by changes in context-dependent fitness effects of hosting the bacteria [31]. In contrast to other studies, the dynamics of *Rickettsia* in this study were observed in a shorter period of time and could have been driven by specific environmental factors.

The facultative endosymbionts’ phylogenetic trees revealed a very homogeneous genetic background for both *Hamiltonella* and *Rickettsia* in Florida (Figure 3). It is already known that *Hamiltonella* sequences of whiteflies from different parts of the world have low genetic diversity [9] and our data shows that populations from Florida are no exception. This study also classified *Rickettsia* strains from different parts of the world into three main groups [9]. The *Rickettsia* strain found in Florida belongs to the clade R1, which consists of invasive and indigenous members from Australia, China, Japan, Israel and Sub-Saharan African countries. A recent study suggested that the majority of the endosymbionts were acquired before the start of *B. tabaci* complex speciation and their spread occurred after this speciation [9]. The characterization of the *Rickettsia* strain found in Florida is helpful in a global context for understanding the complexity and diversity of the facultative endosymbionts in *B. tabaci*.

## 5. Conclusions

The only facultative endosymbionts found among the *B. tabaci* populations collected in Florida were *Hamiltonella* and *Rickettsia*. The phylogenetic analysis revealed low genetic diversity of whiteflies and their endosymbionts. Further studies are being carried out to verify if there is any evidence of the association between the set of facultative endosymbionts in whiteflies and pest status in the field, such as insecticide resistance.

## Figures and Tables

**Figure 1 insects-11-00179-f001:**
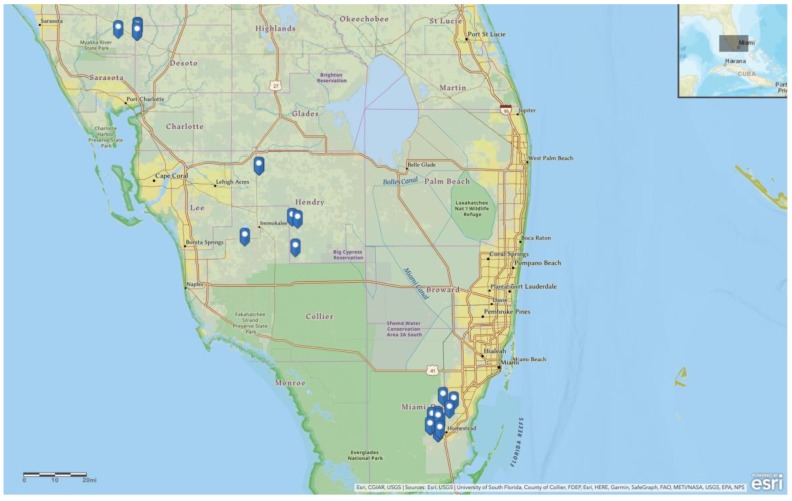
Blue icons represent the collection sites of the 27 *Bemisia tabaci* field populations surveyed in the main tomato growing areas of Florida during 2019. The map was generated by the ArcGIS mapping software. Map sources: Esri, CGIAR, USGS, University of South Florida, County of Collier, FDEP, HERE, Garmin, SafeGraph, FAO, METI/NASA, EPA, NPS.

**Figure 2 insects-11-00179-f002:**
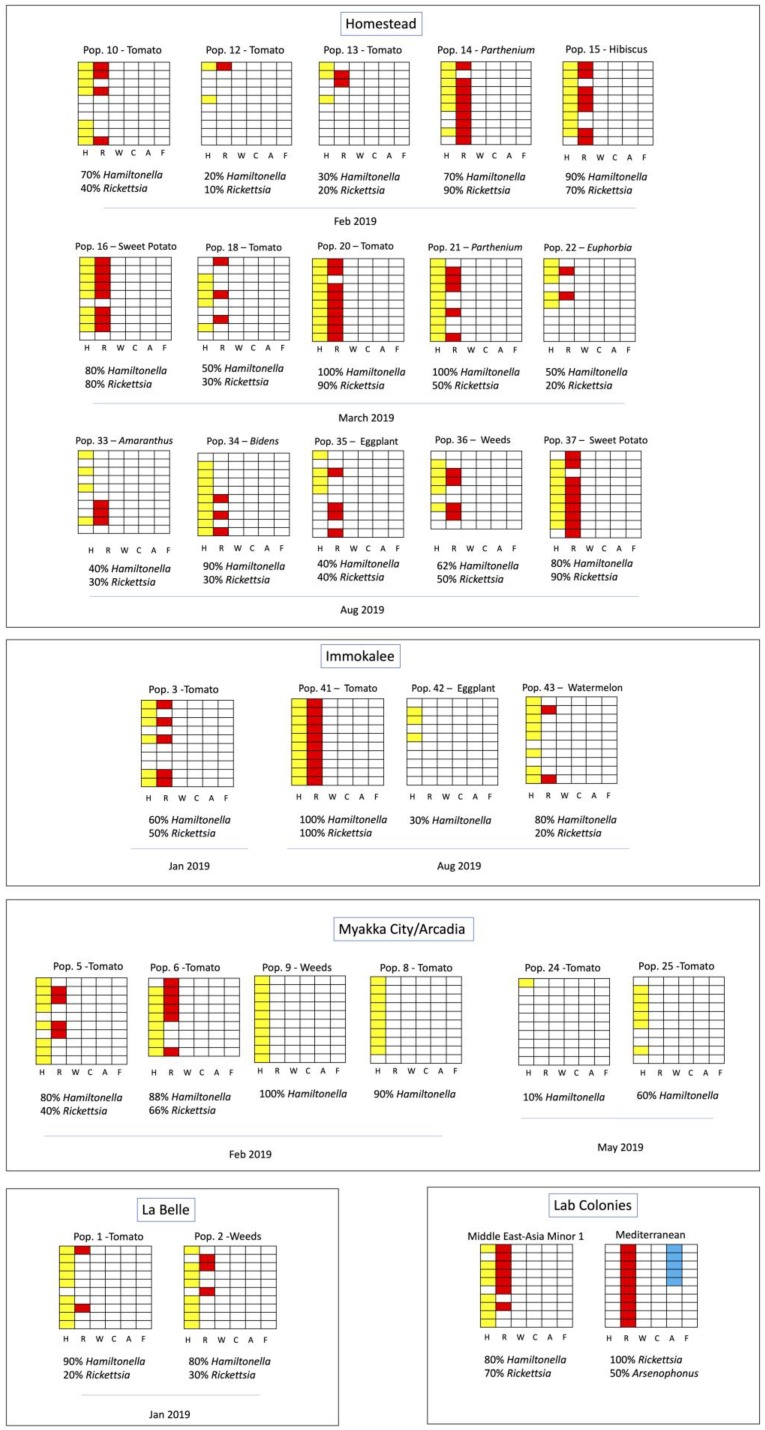
Set of facultative endosymbionts from *Bemisia tabaci* populations collected in Florida. (H) *Hamiltonella*, (R) *Rickettsia*, (W) *Wolbachia*, (C) *Cardinium*, (A) *Arsenophonus*, (F) *Fritschea*. Vertical columns represent the different symbionts tested as indicated by the letters at the bottom of each column, and each horizontal column represents one individual tested for the presence of the six facultative endosymbionts. Positive infections are represented by color filled columns, yellow (*Hamiltonella*), red (*Rickettsia*) and blue (*Arsenophonus*). Collection site, date, plant host and population IDs are indicated.

**Figure 3 insects-11-00179-f003:**
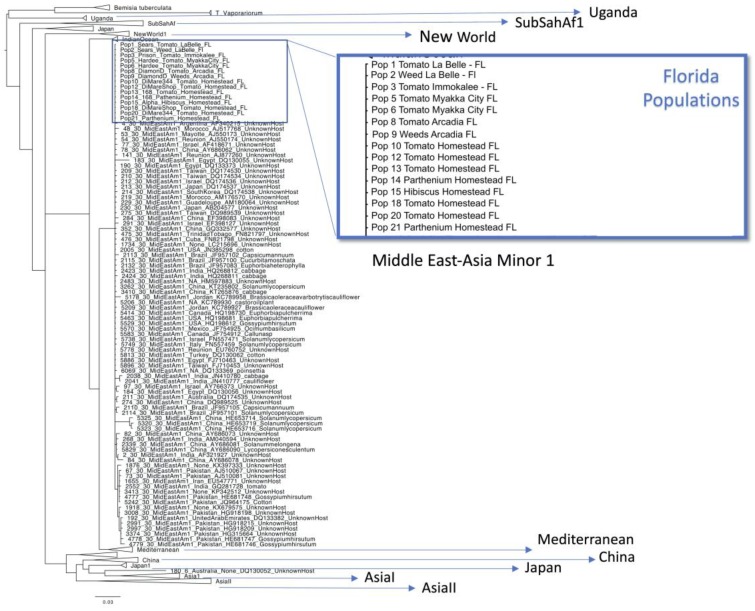
Phylogenetic analysis of the mtCOI gene from *Bemisia tabaci* populations collected in Florida.

**Figure 4 insects-11-00179-f004:**
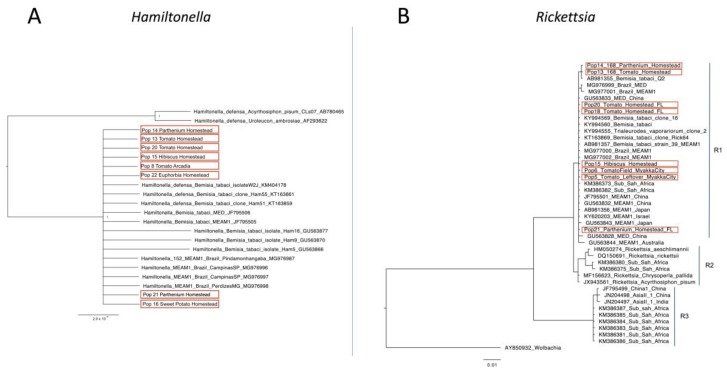
Phylogenetic analysis of the 16S gene rDNA of the facultative endosymbionts *Hamiltonella* (**A**) and *Rickettsia* (**B**) from *Bemisia tabaci* populations. The samples from Florida, USA, obtained in this study are highlighted in red boxes.

**Table 1 insects-11-00179-t001:** Populations of *Bemisia tabaci* collected in Florida and their set of endosymbionts.

Pop. ID	Collection Date	Town	Host Crop	Species	*Hamiltonella* GenBank Accession	*Rickettsia* GenBank Accession	mtCOI GenBank Accession	Endosymbionts					
								H	R	W	C	A	F
B Colony	1990s	Bradenton	*Gossypium hirsutum*	MEAM1	-	-	-	8/10	7/10	0/10	0/10	0/10	0/10
Q Colony	Jul 2017	Palm Beach	*G. hirsutum* *	MED	-	-	-	0/10	10/10	0/10	0/10	5/10	0/10
1	Jan 2019	LaBelle	*Solanum lycopersicum*	MEAM1	-	-	MN972494	9/10	2/10	0/10	0/10	0/10	0/10
2	Jan 2019	LaBelle	Weeds	MEAM1	-	-	MN972495	8/10	3/10	0/6	0/10	0/10	0/10
3	Jan 2019	Immokalee	*S. lycopersicum*	MEAM1	-	-	MN972496	6/10	4/10	0/10	0/10	0/10	0/10
5	Jan 2019	Myakka City	*S. lycopersicum*	MEAM1	-	MN973880	MN972497	8/10	4/10	0/10	0/10	0/10	0/10
6	Feb 2019	Myakka City	*S. lycopersicum*	MEAM1	-	MN973881	MN972498	8/9	6/8	0/9	0/10	0/10	0/10
8	Feb 2019	Arcadia	*S. lycopersicum*	MEAM1	MN973892	-	MN972508	9/10	0/10	0/10	0/10	0/10	0/10
9	Feb 2019	Arcadia	Weeds	MEAM1	-	-	MN972499	10/10	0/10	0/10	0/10	0/10	0/10
10	Feb 2019	Homestead	*S. lycopersicum*	MEAM1	-	-	MN972500	7/10	4/10	0/10	0/10	0/10	0/10
12	Feb 2019	Homestead	*S. lycopersicum*	MEAM1	-	-	MN972501	2/7	1/10	0/10	0/10	0/10	0/10
13	Feb 2019	Homestead	*S. lycopersicum*	MEAM1	MN973889	MN973886	MN972502	3/10	2/10	0/10	0/10	0/10	0/10
14	Feb 2019	Homestead	*Parthenium* sp.	MEAM1	MN973888	MN973887	MN972503	7/10	9/10	0/10	0/10	0/10	0/10
15	Feb 2019	Homestead	*Hibiscus* sp.	MEAM1	MN973891	MN973882	MN972504	9/10	7/10	0/10	0/10	0/10	0/10
16	Mar 2019	Homestead	*Ipomoea batatas*	MEAM1	MN973895	-	-	8/10	8/10	0/10	0/10	0/10	0/10
18	Mar 2019	Homestead	*S. lycopersicum*	MEAM1	-	MN973883	MN972505	5/10	3/10	0/10	0/10	0/10	0/10
20	Mar 2019	Homestead	*S. lycopersicum*	MEAM1	MN973890	MN973884	MN972506	10/10	9/10	0/10	0/10	0/10	0/10
21	Mar 2019	Homestead	*Parthenium* sp.	MEAM1	MN973894	MN973885	MN972507	10/10	5/10	0/10	0/10	0/10	0/10
22	Mar 2019	Homestead	*Euphorbia* SP.	MEAM1	MN973893	-	-	5/10	2/10	0/10	0/10	0/10	0/10
24	May 2019	Myakka City	*S. lycopersicum*	MEAM1	-	-	-	1/10	0/10	0/10	0/10	0/10	0/10
25	May 2019	Myakka City	*S. lycopersicum*	MEAM1	-	-	-	6/10	0/10	0/10	0/10	0/10	0/10
33	Aug 2019	Homestead	*Amaranthus* sp.	MEAM1	-	-	-	4/10	3/10	0/10	0/10	0/10	0/10
34	Aug 2019	Homestead	*Bidens* sp.	MEAM1	-	-	-	9/10	3/10	0/10	0/10	0/10	0/10
35	Aug 2019	Homestead	*Solanum melongena*	MEAM1	-	-	-	4/10	4/10	0/10	0/10	0/10	0/10
36	Aug 2019	Homestead	Weeds	MEAM1	-	-	-	5/8	4/8	0/10	0/10	0/10	0/10
37	Aug 2019	Homestead	*I. batatas*	MEAM1	-	-	-	8/10	9/10	0/10	0/10	0/10	0/10
41	Aug 2019	Immokalee	*S. lycopersicum*	MEAM1	-	-	-	10/10	10/10	0/10	0/10	0/10	0/10
42	Aug 2019	Immokalee	*S. melongena*	MEAM1	-	-	-	3/10	0/10	0/10	0/10	0/10	0/10
43	Aug 2019	Immokalee	*Citrullus lanatus*	MEAM1	-	-	-	8/10	2/10	0/10	0/10	0/10	0/10

* Originally collected from hibiscus in the field but has been maintained on cotton since July 2017.

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
