# Peer review of "Bacterial Endosymbiont Diversity among Bemisia tabaci (Hemiptera: Aleyrodidae) Populations in Florida"

_insects, 2020, doi:10.3390/insects11030179_

Round 1

Reviewer 1 Report

The authors have presented a survey of bacterial endosymbiont diversity in Bemisia tabaci (Hemiptera), from surveying a few counties in South Florida. The manuscript has laid out the results of their survey; however the authors need to improve upon the interpretation and summarization of their results. Follow up studies are required to make some of the conclusions drawn in the discussions. Please see some suggestions on improving this manuscript;

Abstract:

Lines 9-10:“The sweetpotato whitefly, Bemisia tabaci (Hemiptera: Aleyrodidae), is a pest of many agricultural crops and an important vector of economically important plant viruses.” Please reword the sentence to highlight the economic importance of the crops, and not the plant viruses.

Introduction

This section should lay out clearly the crops on which the B tabaci are found within South Florida. Please provide more details in this section.

Line 48-49: Can you please elaborate on any hypothesis regarding the role of endosymbionts in B tabaci?

Methods:

Section 2.1:

Lines 52-55: A collection map of the white flies would be required in this section. Also please specify the crop the white fly was collected from.

Section 2.2:

Lines 65-69: Why was the modified Chelex method chosen for extraction over other genomic DNA extraction methods? Please elaborate.

Lines 74-76: Please clarify that sanger sequencing of the mtCOI fragment was carried out at Genewiz (along with the location of the facility).

Lines 79-83: Can you clarify why only some of the representative populations were sequenced? Did you sequence every endosymbiont? Did you compare this sequence to the sequence available on public databases? Please add this information to this section.

Lines 90-97: The detailed numbers belong in the results section. You can explain the methods without delving too much into results in this section.

Results:

Figure 1: What do the red, yellow and blue boxes represent? What does the matrix represent? Please explain either in the figure description or in a legend within the figure.

Figure2, 3a & 3b: It is very hard to decipher anything at the resolution provided. Can you please re export your phylogenetic tree and import it back into the document to improve the resolution.

Discussions:

Is there any reason from for sampling only the 5 chosen counties? Do they represent the whole of South Florida? A map in the introduction section with the sampling locations, crops found in the field, and your classification of southern Florida, would be extremely useful in this regard.

Lines 143-144: Can you please elaborate on context driven fitness effects of hosting the bacteria? The reason for decline in Rickettsia within the same field in a three month span is still not clear to me. Please also elaborate on the other studies you have mentioned where they have observed the same trend but over a longer time span. Perhaps it will be better to follow this observation for a longer time period to document shifts or increase or decrease in abundance of different endosymbionts.

Lines 149-151: Since not everyone is familiar with the documented history of introductions of Rickettsia R1 strain from around the world, Can you please provide a brief statement or two elaborating on this reference?

Author Response

Abstract:

We appreciate the comments and all the changes requested were performed. Point-by-point response is found below.

Lines 9-10:“The sweetpotato whitefly, Bemisia tabaci (Hemiptera: Aleyrodidae), is a pest of many agricultural crops and an important vector of economically important plant viruses.” Please reword the sentence to highlight the economic importance of the crops, and not the plant viruses. Reply: Done

Introduction

This section should lay out clearly the crops on which the B tabaci are found within South Florida. Please provide more details in this section. Reply: Done

Line 48-49: Can you please elaborate on any hypothesis regarding the role of endosymbionts in B tabaci? Reply: The knowledge about the whitefly vector and its facultative endosymbiont is important for better understanding the epidemics of whitefly-transmitted viruses and can be helpful in resistance monitoring studies as well. The data provided in this research will supplement further studies to find a connection between facultative endosymbionts and pest status.

Methods:

Section 2.1:

Lines 52-55: A collection map of the white flies would be required in this section. Also please specify the crop the white fly was collected from. Reply: The map and the sampled crops have been added to the manuscript.

Section 2.2:

Lines 65-69: Why was the modified Chelex method chosen for extraction over other genomic DNA extraction methods? Please elaborate. Reply: The Chelex method is one of the simplest, cheapest and most efficient methods for DNA extraction of whiteflies. It has been used in several publications in the past years and is very well optimized in the lab where the research was carried out.

Lines 74-76: Please clarify that sanger sequencing of the mtCOI fragment was carried out at Genewiz (along with the location of the facility). Reply: Done

Lines 79-83: Can you clarify why only some of the representative populations were sequenced? Did you sequence every endosymbiont? Did you compare this sequence to the sequence available on public databases? Please add this information to this section. R: We only sequenced the facultative endosymbionts that were found in the field populations (Hamiltonella and Rickettsia). We tried to sequence at least one individual per collection site to represent all the area where the survey was carried out. However, not all the sequencing data had good quality (most likely because of DNA concentration and DNA purity issues). Thus, only high-quality sequencing data were included in the phylogenetic analysis. The data was compared to the NCBI database. We added the information to the manuscript accordingly.

Lines 90-97: The detailed numbers belong in the results section. You can explain the methods without delving too much into results in this section. Reply: The methods were rephrased and some sentences moved to the results section as suggested.

Results:

Figure 1: What do the red, yellow and blue boxes represent? What does the matrix represent? Please explain either in the figure description or in a legend within the figure. Reply: Done

Figure2, 3a & 3b: It is very hard to decipher anything at the resolution provided. Can you please re export your phylogenetic tree and import it back into the document to improve the resolution. Reply: Figures with high resolution (300 dpi) were provided in pdf format and can be found at the supplemental file (Figures.zip) for better visualization.

Discussions:

Is there any reason from for sampling only the 5 chosen counties? Do they represent the whole of South Florida? A map in the introduction section with the sampling locations, crops found in the field, and your classification of southern Florida, would be extremely useful in this regard. Reply: The collection sites were chosen because they are the main tomato growing regions in Florida. A map has been added as suggested by the reviewer.

Lines 143-144: Can you please elaborate on context driven fitness effects of hosting the bacteria? The reason for decline in Rickettsia within the same field in a three month span is still not clear to me. Please also elaborate on the other studies you have mentioned where they have observed the same trend but over a longer time span. Perhaps it will be better to follow this observation for a longer time period to document shifts or increase or decrease in abundance of different endosymbionts. Reply: More information about the effects that the presence of Hamiltonella and Rickettsia in the whitefly vector have been added to the discussion. We agree with the reviewer that monitoring over a longer time period would be more appropriated to confirm the dynamics of Rickettsia in a population. However, the sudden decrease of endosymbionts observed in this population over a short time period is also interesting to highlight as it could have been driven by specific environmental factors.

Lines 149-151: Since not everyone is familiar with the documented history of introductions of Rickettsia R1 strain from around the world, Can you please provide a brief statement or two elaborating on this reference? Reply: A recent study suggested that the majority of the symbionts were acquired before the start of B. tabaci complex speciation and their spread occurred after this speciation (Ghanim et al., 2019; doi.org/10.1371/journal.pone.0213946). The characterization of the Rickettsia strain found in Florida is helpful in a global context for understanding the complexity and diversity of the facultative endosymbionts.

Reviewer 2 Report

The manuscript “Bacterial endosymbiont diversity among Bemisia tabaci (Hemiptera: Aleyrodidae) populations in Florida” by De Marchi and Smith describes the facultative endosymbiont species associated with the whitefly populations from Florida.

The ms is well written and methodology seems correct to me (although I am not an endosymbiont specialist, the methods used are scientifically sound).

My main concern is on the very poor level of novelty of this contribution. Many papers have been published on B. tabaci endosymbionts, and most of them provided valuable results in term of symbiont identification and characterization, interaction/correlation between symbionts and virus transmission or insecticide resistance, pattern of symbiont inheritance to the progeny and more. This manuscript is a very simple description of the symbiont species associated with some populations of B. tabaci from a restricted area (Florida). The two symbiont species found are already known for B. tabaci and no scientific hypotheses are proven in such work, which is merely descriptive. According to this, the discussion section is very poor, as the work is not aimed at testing biological/evolutionistic/functional hypotheses. In my opinion this contribution is not worthy of publication in an international journal because it adds very little information and it is not clear why the work has been done.

Author Response

We understand the concern of the reviewer. However, we would like to point out that Florida is the second producer of tomatoes in the United States and whiteflies and whitefly-transmitted viruses are the major concerns for the tomato crop in the State. Currently, there is no information available in the literature about whitefly facultative endosymbionts in Florida. In recent years, several publications highlighted the importance of facultative endosymbionts and their association with virus transmission efficiency, insecticide resistance, and thermotolerance. The first step in our research is to observe the patterns of endosymbionts across Florida and subsequently we will perform assays to evaluate insecticide resistance with the same populations. We believe that the information obtained in this study, which includes whitefly species identification/phylogenetic analysis and endosymbionts identification/phylogenetic analysis, is essential and will supplement further studies regarding whitefly management in Florida.

Round 2

Reviewer 1 Report

The manuscript has improved significantly with the changes made by the authors. One minor change that I would request is that a map generated by ArcGIS or DIVA-GIS be used in the manuscript as opposed to the one generated by google maps. 

Thanks

Author Response

We appreciate the comments. Figure 1 was replaced by a new map generated by ArcGIS as suggested. The high-resolution picture can be found in the "Figures.zip" file.

Author Response

We appreciate the comments.